# Management of a Case of Peritonitis Due to *Neisseria gonorrhoeae* Infection Following Pelvic Inflammatory Disease (PID)

**DOI:** 10.3390/antibiotics9040193

**Published:** 2020-04-18

**Authors:** Maria A. De Francesco, Paola Stefanelli, Anna Carannante, Silvia Corbellini, Cinzia Giagulli, Giovanni Lorenzin, Maurizio Ronconi, Elisa Arici, Monica Cadei, Riccardo Campora, Arnaldo Caruso

**Affiliations:** 1Department of Molecular and Translational Medicine, Institute of Microbiology, University of Brescia-Spedali Civili, 25123 Brescia, Italy; silvia.corbellini@gmail.com (S.C.); cinzia.giagulli@unibs.it (C.G.); giovanni.lorenzin@hotmail.it (G.L.); arnaldo.caruso@unibs.it (A.C.); 2Department Infectious Diseases, Istituto Superiore di Sanità, 00161 Rome, Italy; paola.stefanelli@iss.it (P.S.); anna.carannante@iss.it (A.C.); 3General Surgery, ASST- Spedali Civili of Gardone Val Trompia and Brescia, 25063 Brescia, Italy; maurizio.ronconi@asst-spedalicivili.it (M.R.); elisa.arici90@gmail.com (E.A.); 4Radiology Unit, ASST-Spedali Civili of Gardone Val Trompia, 25063 Brescia, Italy; dott.cadei@gmail.com; 5Department of Radiology, University and ASST Spedali Civili of Brescia, 25063 Brescia, Italy; camporic@hotmail.it

**Keywords:** PID, acute abdomen, laparascopy, *Neisseria*, *Chlamydia*

## Abstract

Pelvic inflammatory disease (PID), a serious infection in sexually active women, is one of the reasons for which females seek care in emergency departments and therefore represents an important public health problem. PID is the result of an endocervical infection with different microorganisms, which then ascend to the endometrium and fallopian tubes. Symptoms of PID may be mild and aspecific, making its diagnosis difficult. However, this clinical condition requires effective antibiotic treatment to reduce incidence of complications and late sequelae. We describe here a case of peritonitis as a complication of pelvic inflammatory disease (PID) due to *Neisseria gonorrhoeae* infection in a 49-year-old woman who presented at the Emergency Department with acute abdominal pain.

## 1. Introduction

Pelvic inflammatory disease (PID) is an acute infection of the upper genital tract involving all neighboring structures as a consequence of ascending infections from the lower genital tract. Bacteria spread from the vagina to the cervix, to endometrium, and then onto the upper genital tract [1]. No specific international data are available for PID incidence worldwide. In 2005, the World Health Organization (WHO) estimated that about 448 million new cases of sexually transmitted diseases (STIs) occur in subjects aged 15–49 years [2]. The problem of determining the worldwide incidence of PID is linked both to the lack of specific symptoms of primary pelvic inflammation and to a clinical resolution by the time of presentation. Common clinical symptoms, in fact, include pelvic pain, fever, vomiting, dyspareunia, leukocytosis, and other vague constitutional symptoms. PID requires effective treatment to reduce incidence of complications and late sequelae including tubal damage, uterine and tubo-ovarian abscess, pelvic peritonitis, development of abdominal and pelvic peritoneal adhesion, and perihepatitis (Fitz–Hugh–Curtis syndrome).

*Chlamydia trachomatis* and *Neisseria gonorrhoeae* were considered the main pathogens in the etiology of PID [1], even if other microorganisms such as *Mycoplasma genitalium* and anaerobic bacteria were also involved [3,4]. Furthermore, polymicrobial infections were responsible of 30–40% of PID cases [5].

Here, we report the case of a woman with a diffuse peritonitis as a complication of pelvic inflammatory disease due to *N. gonorrhoeae* infection.

## 2. Case Study

A 49-year-old woman attended at the Emergency Department of ASST-Spedali Civili of Gardone Val Trompia, Brescia, Italy in April 2019 with acute abdominal pain at central upper quadrants starting 24 h before. Clinical examination revealed tenderness at superficial palpation. She denied being pregnant and had no history of chronic disease or previous abdominal trauma or surgery. She was febrile with a body temperature of 38.5 °C. On physical examination, she had a blood pressure of 115/60 mmHg and a heart rate of 103 bpm. Chest and abdominal X-ray studies performed on hospital admission were normal. Laboratory tests showed raised inflammatory markers with total WBC count of 34.310 mm^3^ and CRP level of 161 mg/L, and a mild microcytic anemia (Hb 11.2 g/dL, MCV 69.2 fl). Liver enzymes, bilirubin, albumin, and prothrombin time were normal. Urinalysis was normal. Because of the persistent pain, the resistance to standard analgesic drugs and the presence of inflammatory markers in blood, a CT scan of the abdomen and pelvis with intravenous contrast was performed. It revealed dilated loops of proximal small bowel with wall thickening and an air fluid level (Figure 1A,B).

There was a normal hepato-biliary structure with no evidence of hepatic abscess, acute cholecystitis, or dilation of the intra or extra-hepatic bile ducts. There was no radiologic suspicion of PID (Figure 1C). The patient was therefore transferred to general surgery with a diagnosis of acute abdomen and a possible bowel occlusion. For this reason, an exploratory laparoscopy was performed, which evidenced free non-malodorous pus in the pelvis. The intestinal mass showed signs of peritonitis and minimal distension as in the paralytic ileum, without significant changes of caliber or adhesions. The uterus appeared edematous, erythematosus, and swollen. The diagnosis of severe pelvic inflammatory disease was made. After collecting peritoneal liquid for bacterial culture, a peritoneal lavage was performed and an antibiotic therapy was immediately started (azithromycin and metronidazole, 500 mg each daily, because the patient was allergic to beta-lactams, for 14 days).

Bacterial culture was positive for gram negative diplococci. MALDI-TOF MS analysis allows to identify *N. gonorrhoeae*.

The *N. gonorrhoeae* isolate was stored at −80 °C in brain heart infusion medium (Oxoid, Ltd., Milan, Italy) containing 20% glycerol and sent to the National Reference Laboratory at Istituto Superiore di Sanità for phenotypic and genotypic characterization. Antimicrobial susceptibility was performed after growth on Thayer–Martin medium (Oxoid Ltd., Italy) with 1% IsoVitalex (Oxoid Ltd., Italy) at 37 °C in a 5% CO2 atmosphere, following the Euro-GASP (European Gonococcal Antimicrobial Surveillance Programme) recommendations [6]. The MIC TEST STRIP method (Liofilchem Diagnostici, Teramo, Italy) was carried out in agreement with the manufacturer’s instructions to determine the antimicrobial susceptibility against cefixime, ceftriaxone, ciprofloxacin, azithromycin, and spectinomycin (Table 1).

The minimum inhibitory concentration (MIC) values refer to resistant EUCAST clinical breakpoint (version 9.0, 2019), as follows: cefixime and ceftriaxone >0.125 mg/L; ciprofloxacin >0.06 mg/L; azithromycin >1 mg/L [7].

The World Health Organization (WHO) *N. gonorrhoeae* G, K, M, O, and P control strains were used in each assay [8].

The *β*-lactamase production to identify Penicillinase Producing *N. gonorrhoeae* (PPNG) isolates was performed by the chromogenic test using Beta Lactamase Test (Liofilchem Diagnostici, Italy).

Sequencing of *porB* and *tbpB* genes of NG-MAST (*Neisseria gonorrhoeae* Multiantigen Sequence Typing) analysis was obtained as previously described [9]. The *porB* and *tbpB* alleles and the sequence type (ST) were assigned according to the NG-MAST website (www.ng-mast.net), following the interpretative procedures; the genogroup (G) was determined referring to the European Centre for Disease Prevention and Control (ECDC) definition [10].

*GyrA* and *parcC* genes were amplified and sequencing to investigate ciprofloxacin resistance [11].

The gonococcal isolate was PPNG-negative and susceptible to all the antimicrobials tested except to ciprofloxacin (MIC value of 12 mg/L). It belonged to the ST-2400. This sequence type is part of the genogroup G2400 associated in Europe, and also in Italy, to gonococcal ciprofloxacin resistant isolates [10,12]. The *gyrA* gene showed mutations that encoded the amino acid substitutions D91F and D95G, whereas the *parC* gene was wild-type.

Molecular confirmation by PCR was performed after 14 days, resulting negative for *N. gonorrhoeae.*

The patient was discharged after 20 days in good condition and with resolution of all symptoms without signs of recurrence at the follow-up.

## 3. Discussion

Acute abdominal pain represents a clinical and radiologic problem. Many patients with a primary diagnosis of bowel obstruction or appendicitis show successively to have PID. The diagnosis of PID is often difficult when a patient presents to an emergency department with non-specific symptoms only. The accumulation of inflammatory exudates in peritoneum causes the sharp right upper quadrant abdominal pain. This pain might be more intense with deep respiration or coughing due to negative pressure during respiratory movements. PID can be misdiagnosed because it is similar to many other disorders such as pulmonary embolism, pneumonia, cholecystitis, renal colic, and perforated ulcer. Diagnosis is quite difficult because there are aspecific physical signs, so radiographic studies are important to exclude other causes, and laboratory tests are commonly negative or only slightly elevated such as electrolytes, liver function test, and white blood cell count [13]. Because there is no specific test, diagnosis of PID is generally clinical and combines medical history, bimanual examination to elicit cervical motion tenderness and adnexal or uterine tenderness, and pelvic ultrasound. So far, microbiological confirmation by positive NAATs for *C. trachomatis*, *N. gonorrhoeae*, or *M. genitalium* is very important. Our case is quite similar to other sporadic cases reported in the literature [12,14,15,16], where women 30 years of age were admitted to emergency departments with acute abdominal pain and a diagnosis of peritonitis was made only after laparascopy with isolation of *N. gonorrhoeae* from peritoneal liquid. 

This further case report may alert clinicians to have the suspicion about the possible involvement of ascending genital infections in sexually active women with unknown causes of abdominal pain. This awareness can lead to an early diagnosis and to a correct management of such patients. Because this clinical picture is quite confusing, microbiological analysis is essential for diagnosis and for guiding the correct antimicrobial that resolve the symptoms and prevent the long-term complications associated with PID.

## Figures and Tables

**Figure 1 antibiotics-09-00193-f001:**
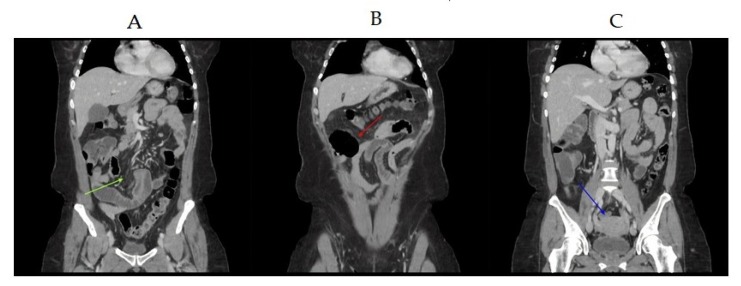
Abdominal computed tomography showing a loop of the small bowel (green arrow) (**A**) with a mild retrograde dilatation and an air-fluid level (red arrow) (**B**). No radiographic evidence of pelvic inflammatory disease was observed; in particular, no abscesses or effusion were detected. The uterus appears morphologically normal (blue arrow) (**C**).

**Table 1 antibiotics-09-00193-t001:** Antimicrobial minimum inhibitory concentrations (MICs) of *N. gonorrhoeae* isolate.

Antimicrobial	MIC Value (mg/L)
Cefixime	0.047
Ceftriaxone	0.016
Ciprofloxacin	12
Azithromycin	1
Spectinomycin	12

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
