# Peer review of "Management of a Case of Peritonitis Due to Neisseria gonorrhoeae Infection Following Pelvic Inflammatory Disease (PID)"

_antibiotics, 2020, doi:10.3390/antibiotics9040193_

Round 1

Reviewer 1 Report

The authors did a fine job describing their case report of peritonitis linked to PID. The work provides an additional example for the scientific literature that will help clinicians understand symptoms associated with PID complications and what was accomplished to diagnose and treat the infection effectively.

Some minor English errors are present but they do not impede clear communication of the findings.

For example, line 46's sentence "A 49-years old woman attended at Emergency Department of ASST-Spedali Civili of Gardone Val Trompia, Brescia, Italy in April 2019, with an acute abdominal pain . . ." would read better if changed to "A 49-year old woman presented at the Emergency Department of ASST-Spedali Civili of Gardone Val Trompia, Brescia, Italy in April 2019, with acute abdominal pain . . ."

Lines 17-18 of the abstract read better to me if written as such:PID is the result of an endocervical infection with different microorganisms, which then ascend to the endometrium and fallopian tubes.

Author Response

All the minor English errors indicated by the reviewer have been corrected.

Point 1: A 49-years old woman attended at Emergency Department of ASST-Spedali Civili of Gardone Val Trompia, Brescia, Italy in April 2019, with an acute abdominal pain

Response 1: A 49-year old woman presented at the Emergency Department of ASST-Spedali Civili of Gardone Val Trompia, Brescia, Italy in April 2019, with acute abdominal pain

Point 2: PID is the result of  the endocervix infection with different microorganisms, which then  arrive to the endometrium and fallopian tube

Response 2: PID is the result of anendocervical infection with different microorganisms, which then ascend to the endometrium and fallopian tubes

Reviewer 2 Report

Overall, this is a straight forward clinical case report of a rare complication to gonococcal infection.  The only suggestion might be to include the MIC data in a table, as this would make the data more valuable to the research community.

Author Response

Point 1: The only suggestion might be to include the MIC data in a table, as this would make the data more valuable to the research community.

Response 1: The Table has been inserted

Reviewer 3 Report

This case report, entitled ”Management of a case of peritonitis due to Neisseria gonorrhoeae infection following pelvic inflammatory disease (PID)” was structured properly. I have simple suggestions.

  1. in case study section

Pelvic exam should be described. Bimanual exam to evaluate cervix motion pain or adnexa tenderness not only radiologic suspicion of PID.

  1. in discussion section

Diagnosis of PID should be discussed, including clinical approach to the diagnosis of PID.

Author Response

Point 1: in case study section

Pelvic exam should be described. Bimanual exam to evaluate cervix motion pain or adnexa tenderness not only radiologic suspicion of PID.

Response 1: Pelvic exam has not been performed because there was no suspicion of PID and lack of gynaecological symptoms, in fact the patient was treated for a possible bowel occlusion

Point 2: in discussion section

Diagnosis of PID should be discussed, including clinical approach to the diagnosis of PID

Response 2: Diagnosis of PID has been inserted in the Discussion section from line 137 to 139.